# Antibiotic Resistance of *Staphylococcus aureus* Strains—Searching for New Antimicrobial Agents—Review

**DOI:** 10.3390/ph18010081

**Published:** 2025-01-11

**Authors:** Michał Michalik, Adrianna Podbielska-Kubera, Agnieszka Dmowska-Koroblewska

**Affiliations:** MML Medical Centre, Bagno 2, 00-112 Warsaw, Poland; michal.michalik@mml.com.pl (M.M.); agnieszkadmowska@wp.pl (A.D.-K.)

**Keywords:** *Staphylococcus aureus*, MRSA, multidrug resistance, methicillin resistance, phages, antimicrobial peptides, alternative therapies, natural therapies

## Abstract

Inappropriate and excessive use of antibiotics is responsible for the rapid development of antimicrobial resistance, which is associated with increased patient morbidity and mortality. There is an urgent need to explore new antibiotics or alternative antimicrobial agents. *S. aureus* a commensal microorganism but is also responsible for numerous infections. In addition to innate resistance to β-lactam antibiotics, *S. aureus* strains resistant to methicillin (MRSA) often show resistance to other classes of antibiotics (multidrug resistance). The advancement of phage therapy against MRSA infections offers a promising alternative in the context of increasing antibiotic resistance. Therapeutic phages are easier to obtain and cheaper to produce than antibiotics. However, there is still a lack of standards to ensure the safe use of phages, including purification, dosage, means of administration, and the quantity of phages used. Some bacteria have developed defense mechanisms against phages. The use of phage cocktails or the combination of antibiotics and phages is preferred. For personalized therapy, it is essential to set up large collections to enable phage selection. In the future, the fight against MRSA strains using phages should be based on a multidisciplinary approach, including molecular biology and medicine. Other therapies in the fight against MRSA strains include the use of endolysin antimicrobial peptides (including defensins and cathelicidins). Researchers’ activities also focus on the potential use of plant extracts, honey, propolis, alkaloids, and essential oils. To date, no vaccine has been approved against *S. aureus* strains.

## 1. Staphylococci

The inappropriate and excessive use of antibiotics is responsible for the rapidly increasing antimicrobial resistance. The World Health Organization has recognized this resistance as a widespread threat to human health, classifying it as one of the top 10 most serious health threats [1]. The incidence of antimicrobial resistance increased significantly during the COVID-19 pandemic. A study published in the *Lancet* journal confirmed that there were approximately 1.27 million deaths worldwide in 2019 related to antimicrobial resistance in bacteria [2]. It is predicted that by 2050, infections associated with antimicrobial-resistant bacteria (multidrug-resistant bacteria, MDRB) will be associated with approximately 10 million deaths per year, surpassing the mortality attributed to cancer. Furthermore, it is estimated that MDRB will become a major financial burden on the global economy. By 2050, the bacteria could cause an annual loss of USD 120 trillion in US gross domestic production, resulting from the increase in the number of people unable to work due to diseases and the increased number of deaths among farm animals infected with MDRB pathogens [2].

Among the antibiotic-resistant pathogenic bacteria, a serious clinical threat is posed by pathogens defined by the acronym ESKAPEE (*Enterococcus faecium*, *Staphylococcus aureus*, *Klebsiella pneumoniae, Acinetobacter baumannii*, *Pseudomonas aeruginosa*, *Enterobacter* spp., and *Escherichia coli*) [3]. Resistant infections mostly affect patients undergoing long-term hospitalization or treatment with antimicrobial agents, most often from intensive care units, as well as patients after invasive procedures or patients who are immunocompromised, e.g., cancer patients. The growing number of “superbugs” resistant to the most commonly prescribed antibiotics has contributed to the increased number of illnesses and deaths due to infections and the lack of alternative methods of treating them. Scientists’ activities should focus on implementing new alternative antimicrobial therapies in the shortest possible time [1].

Methicillin-resistant *S. aureus* (MRSA), vancomycin-resistant *Enterococcus faecium*, and β-lactamase-resistant *Streptococcus pneumoniae* are among the clinically studied antimicrobial-resistant bacteria [1].

*S. aureus* is a commensal microorganism that colonizes the nasal cavity in approx. 30% of adults. *S. aureus* is also responsible for numerous skin and soft tissue infections, which can spread to other tissues and organs, leading to bacteremia and even severe septic shock, with a mortality rate of up to 25%. *S. aureus* can further contribute to osteomyelitis, arthritis, pneumonia, endocarditis, meningitis, infection of hematopoietic organs, food poisoning, urinary tract and surgical site infection, and toxic shock syndrome [1]. The presence of *S. aureus* strains has also been linked to chronic rhinosinusitis [4]. The *Lancet* documented *that S. aureus* strains are one of the main infectious agents responsible for high mortality in 135 countries. The highest number of deaths related to this bacterium was observed among people over 15 years [2].

The presence of virulence factors and biofilm formation are important features that protect *S. aureus* strains from the host immune system and antimicrobial agents while allowing the bacteria to adhere effectively to host surfaces and colonize preferred anatomical niches. Medical devices implanted in the body are the basis for the formation of biofilms by *S. aureus* strains and the development of infections with high resistance to antimicrobial agents. Biofilms are structures that protect bacteria, creating favorable conditions for growth, despite the lack of nutrients, water, or deficiency of optimal environmental temperature. They also protect bacteria from antimicrobial agents. According to literature data, bacteria that form biofilms are 10–1000 times more resistant to antibiotics than bacteria living in a dispersed form [5]. The ability of bacteria to form biofilms significantly influences the increased resistance of bacteria to antibiotics. It has been confirmed that a significant percentage of MRSA strains producing biofilms (43.3%) are much more often resistant to antibiotics compared to strains that do not have this ability [6,7].

MRSA strains responsible for human infections fall into three categories based on their origin: healthcare-associated MRSA (HA-MRSA), community-associated MRSA (CA-MRSA), and livestock-associated MRSA (LA-MRSA). HA-MRSA strains are endemic in many hospitals. The fact that *S. aureus* can be a commensal bacterium increases the risk of self-infection and/or transmission of the pathogen to patients via healthcare workers’ hands or equipment. The procedure of using mupirocin for decolonization of patients is routine before high-risk surgeries (cardiac and orthopedic surgery) due to financial reasons and the risk of selecting bacterial resistance [8]. A characteristic feature of hospital-acquired HA-MRSA clones is the presence of SCCmec type I–III (staphylococcal cassette chromosome mec) cassettes, which, in addition to the mecA gene, may contain genes determining resistance to other groups of antimicrobial drugs [9]. HA-MRSA are usually among the epidemic multidrug-resistant strains. In the epidemiological classification, healthcare-associated MRSA infections are diagnosed in the first 48 h of hospitalization, and non-hospital MRSA infections are diagnosed later. For healthcare-associated infections, the absence of documented healthcare-associated risk factors is required [10]. CA-MRSA strains can be isolated from asymptomatic individuals. These strains differ in antibiotic resistance and the presence of Panton–Valentine leukocidin compared to HA-MRSA strains. CA-MRSA strains should be considered as a potential source of infection outbreaks in hospitals [8]. The presence of CA-MRSA strains in healthcare facilities and communities is an alarm signal indicating the need to monitor the epidemiological situation [1]. LA-MRSA strains have been detected in both livestock and human companion animals, including pigs, poultry, and cattle, as well as in food of animal origin. Literature data indicate that selected LA-MRSA strains, whose presence has been confirmed in poultry meat, can also colonize humans. These strains can adapt to changing environmental conditions and different hosts. In order to follow the evolution of MRSA acquired from animals in humans, the European Food Safety Authority (EFSA) recommends monitoring food-producing animals [8]. Recent studies have shown high rates of MRSA colonization in humans who are in close contact with animals in their professional or home lives [11].

Currently, clinical strategies for the prevention and treatment of infections with *S. aureus* strains are mainly based on antibiotic therapy. Excessive use of antibiotics has contributed to the increasing drug resistance among *S. aureus* strains [1].

In the 1940s, it was reported that resistance to penicillin used to treat *S. aureus* infections developed as a result of mutations in the blaZ gene. The blaZ gene can be located in the chromosome or on a plasmid, which favors the rapid spread of this resistance mechanism between strains. It is estimated that 80–90% of all *S. aureus* isolates carry a penicillin resistance mechanism [10]. For many years, it was thought that methicillin resistance emerged as a response to the use of the antibiotic for treatment. However, as the results of genomic analyses show, MRSA strains were reported even before the introduction of methicillin for treatment, and the widespread use of natural β-lactams such as penicillin led to their selection. β-Lactamases are enzymes that hydrolyze ester and amide bonds in β-lactam antibiotics (penicillins, cephalosporins, monobactams, and carbapenems). Resistance to β-lactam antibiotics in staphylococci is related to the ability to inhibit the synthesis of the bacterial cell wall. All beta-lactam antibiotics act on PBPs (penicillin-binding proteins). Modifications of PBPs represent one of the mechanisms of resistance to this group of drugs. Microorganisms contribute to the reduction of the affinity to PBPs and the increase of the expression of β-lactamase, which is responsible for the hydrolysis of the β-lactam ring of antibiotics [11]. Production of the PBP2a protein is associated with the presence of the acquired mecA gene, located in a mobile genetic element known as the chromosomal cassette, and prevents β-lactam antibiotics from binding to the cell surface due to reduced affinity [10]. A change in PBP2a results in innate resistance to β-lactam antibiotics [11]. The methicillin resistance phenotype can also be mecB and mecC [10].

Methicillin, the first semi-synthetic β-lactam antibiotic, was introduced into clinical practice in the late 1950s. As early as 1961, the first methicillin-resistant strains, referred to as MRSA, were described [10]. Since then, the global prevalence of MRSA infections has increased dramatically. MRSA infections are associated with higher morbidity and mortality compared with infections with *S. aureus* methicillin-sensitive (MSSA) strains, along with higher treatment costs and prolonged hospitalization [12]. Although the prevalence of recorded hospital-acquired MRSA infections has declined in China, Europe, and the US in recent years, they still constitute an important clinical problem in low-income countries. Studies conducted in the United States show that mortality resulting from MRSA infections remains higher compared to other antibiotic-resistant pathogens [5]. MRSA is a significant risk factor for infection in 60 to 80% of the population in developed countries as a result of nosocomial infections [13]. In 2019, MRSA was responsible for more than 100,000 deaths attributable to antimicrobial resistance. Currently, approximately 30% of hospital-acquired infections are associated with MRSA [1].

MRSA strains, apart from their innate resistance to β-lactam antibiotics, may also demonstrate resistance to antibiotics from other classes: aminoglycosides, macrolides, tetracyclines, and fluoroquinolones, which is referred to as multidrug resistance [14,15]. Multi-resistance contributes to a large extent to the reduction of the number of therapeutic solutions and, in many cases, results in the use of antibiotics characterized by numerous side effects and the increased development of antibiotic resistance. Such antibiotics include, among others, vancomycin and linezolid [16,17]. Vancomycin was, for many years, an effective drug against MRSA because methicillin-resistant *S. aureus* strains remained susceptible to it [18]. In clinical practice, we encounter increasingly more vancomycin-resistant MRSA strains [19]. This feature of MRSA strains has been associated with the acquisition of the vanA gene cluster, which is responsible for initiating the resistance mechanism and changing the structure of the bacterial cell wall, which contributes to the reduction of vancomycin affinity to bacteria [20]. Emerging resistance is a significant clinical problem, increasing the risk of treatment failure and contributing to an increase in the percentage of MRSA infections and, thus, the number of patients and death rates [7]. MRSA isolates with a vancomycin-intermediate *S. aureus* (VISA) phenotype, selected following vancomycin therapy, were first described in the late 1990s [10]. In response to this urgent and alarming situation, the World Health Organization has called on researchers and the medical community to take action to implement innovative clinical approaches to reduce the number of MRSA infections [21]. This strategy should include the introduction of new antimicrobial agents, alternative therapies, and improved measures to prevent bacterial infections [1]. Several new drugs are currently in various stages of testing. Some of them have already been accepted by the Food and Drug Administration and the European Medicines Agency. The most clinically promising antibiotics include new antimicrobials from the families of broad-spectrum lipoglycopeptides (dalbavancin, oritavancin, and telavancin), glycopeptide-cephalosporin antibiotic (TD-1792), cephalosporins (ceftaroline and ceftobiprole), and oxazolidinones [8].

## 2. Phage Therapy

Urinary tract infections, infections of the respiratory tract, skin, bone and bone marrow, prostheses and implants, rhinosinusitis, and sinusitis can recur despite antibiotic treatment. As a result, clinicians are increasingly looking for new clinical therapies for patients infected with antibiotic-resistant strains. One such option is phage therapy [3].

Phage therapy is an experimental treatment for treatment-resistant infections, infections with multidrug-resistant microorganisms, and/or biofilm-associated infections. The use of phage therapy should only be considered in infections with confirmed intolerance or the ineffectiveness of antibiotic therapy. Phage therapy currently includes the treatment of bacterial infections. However, there are emerging data indicating that phage therapy could also be used in the future to eliminate other organisms, such as Aspergillus fungi [3].

Bacteriophages were first used to treat staphylococcal skin infections in 1921. These viral pathogens are characterized by their ability to identify bacteria and replicate them [4]. Based on their life cycle, phages can be divided into lysogenic phages and lytic phages. Lysogenic phages integrate their viral genetic material into the host bacterial genome, where their DNA or RNA is replicated by bacterial chromosomes through normal cell division without cell lysis/splitting. In contrast, lytic phages, whose replication cycle is complex, induce lysis of the host bacteria and can thus mediate horizontal gene transfer [22,23]. Phages that transfer bacterial DNA to a new host are often referred to as transduction phages. Adsorption is the initial and critical stage of infection. Phages can bind bacteria with specific receptors on their surface, regardless of whether they are alive or dead. Phage nucleic acid cannot enter dead host bacteria. The major surface proteins consist of phage-encoded polysaccharide depolymerases. These depolymerases have a significant effect on polysaccharide fragments located in the bacterial cell envelope, including lipopolysaccharides of Gram-negative bacteria or the extracellular matrix that constitutes the structure of biofilms [24]. Once adsorbed on the host bacteria, phages using lysozyme produce a hole in the cell wall. The nucleic acid present in the phage head is then incorporated into the bacterial cell, leaving the protein capsid outside. Once the nucleic acid of the phage enters the bacterial cell, it initiates the production of mRNA through transcription. At the same time, it replicates a significant number of nucleic acids for progeny phages, using its own nucleic acid as a matrix. The synthesized proteins and nucleic acids of the progeny phages combine into fully mature phages. When a set threshold of progeny phage numbers is reached, holes are formed in the inner membrane of the bacteria. The endolysins then pass through the inner membrane and target the peptidoglycan in the bacterial cell wall. This action contributes to the disruption of peptidoglycan bonds in the bacterial cell wall, leading to osmotic lysis of the bacteria and the release of progeny phages. The lysogenic cycle leads to the incorporation of phage DNA into the host genome, which in turn produces prophages that are passed on to the next generation of cells. Environmental disturbances, such as food deficiency or the presence of toxic chemicals, can contribute to the excision of the prophage and the initiation of the lytic cycle [2].

The lytic and lysogenic cycles of the phage are shown in Appendix A.

Phages (bacteriophages) are ubiquitous in the environment and represent one of the most abundant forms of biomass present on Earth [25,26]. Bacteriophages are present in most ecosystems [27,28] due to their ability to infect bacterial hosts in soil [29], water [30], animal and human bodies [31], and sewage [32]. Successful clinical trials using phages to combat infections were conducted in the United States in the 1930s [33]. In the following years, phages were used in the prevention and treatment of mycobacterial dysentery and staphylococcal infections in South America [34,35]. Phages have been used as therapeutic agents for almost 100 years in Georgia and Russia [36]. There are no reports in the literature indicating the occurrence of adverse effects or lack of effectiveness of phage use [37,38]. The future success of phage therapy largely depends on the compliance of phage production with pharmaceutical standards, including a safe phage amplification process. Such standards require establishment and approval by national and international drug agencies. Only then will it be possible to adapt them to the specific biological properties of phages [39]. Optimization of these protocols, including maximization of amplification efficiency and minimization of the release of toxic bacterial metabolites and bacterial components to facilitate subsequent purification steps, is an issue that requires further investigation [40]. The development of phage therapy will be supported by the creation of large collections of bacteriophages with a well-characterized range of activity. It allows such phage banks to be quickly searched and the most active phages against a given clinical strain to be selected [41]. Several commercially available therapeutic bacteriophage preparations, such as Bacteriophage-Stafi lokokovyj, Sextaphag, Staphylococcal Bacteriophage, or Pyobacteriophage contain lytic phages that infect *S. aureus*. Topical use of preparations includes purulent infections of the skin and soft tissues, infections of the respiratory tract, eyes, and urinary tract, or decolonization of the nasal cavity in carriers of infectious agents [42,43,44].

Phages have the ability to reduce the minimum inhibitory concentration (MIC) for some antibiotics [45]. This property of phages has been confirmed by studies. The term phage–antibiotic synergy was first introduced by Comeau et al. The researchers accidentally observed that sublethal concentrations of antibiotics significantly increased the production of lytic phages by bacteria [46]. This phenomenon has been associated with increased biomass and biosynthetic capacity of bacteria in the presence of antibiotics at levels sufficient to inhibit cell division without causing cell death. Studies conducted by Gu Liu et al. have shown that phages have the ability to reduce the MIC value required for antibiotic-resistant bacterial strains [47]. The obtained result is influenced by the type of antibiotic and the equilibrium state, as well as the host microenvironment [2].

Phages are extremely species-specific and should therefore be targeted and tested on clinical isolates from specific patients [1]. Due to their widespread presence and abundance, phages are thought to be involved in approximately 20–40% of bacterial lysis cases [48]. Under natural conditions, phage evolution is influenced by the density and diversity of bacterial populations. Host range is the ability of phages to infect a variety of hosts, indicating a high degree of adaptation. To a large extent, host range is influenced by the availability, diversity, and quality of potential hosts [49]. A wide host range allows phages to infect a larger number of hosts. However, the occurrence of phages with an expanded host range may result in reduced replication rates in new hosts and reduced efficiency in primary hosts [2]. For diverse bacterial populations, taking into account the narrow range of host phages, thousands of phages are necessary to cover the diversity of bacterial pathogens. In personalized phage therapy, the rapid selection of phages for a specific patient is crucial for successful therapy [50].

Phages, like other viruses, can interact with the immune system. Such interactions may lead to an antiphage antibody response, as well as an immunomodulatory effect. Phage-dependent immunomodulatory effects may have therapeutic potential in disorders of the immune system, as well as the treatment of certain non-bacterial infections [51]. The findings suggest that the humoral response elicited by phage therapy may depend on the route of administration, type of phage, duration of therapy, and immunocompetence of the patient. The ability of the organism to produce antibodies against intravenously administered phages is used in the diagnosis and monitoring of the health status of patients with immunodeficiency syndrome. The detection of antiphage antibodies in the blood may indicate the regeneration of the immune system after the phage therapy [52].

Phages and bacteria evolved together over billions of years [53]. As a result, bacteria have developed complex defenses against phages. These mechanisms include receptor changes to prevent phage from attaching to the host surface, a failed replication cycle or “suicide death” to rescue neighboring bacteria, restriction-modification systems to degrade phage DNA and prevent replication, and the bacterial RNA interference system, defined as Clustered Regularly Interspaced Short Palindromic Repeats, CRISPR [54]. As a countermeasure, phages have developed various ways to cross bacterial defense lines, including having several receptor-binding proteins and anti-CRISPR genes and using modified nucleotides in their DNA [55]. In practice, the use of phages for therapeutic purposes requires the continuous isolation of phages with novel host specificity.

The commercial development of new antibiotics is a costly and time-consuming process, often with limited cost-effectiveness for industry [7]. The costs of activities related to the development of new antibiotics may amount to as much as USD 1.5 billion [56]. The estimated costs discourage pharmaceutical companies from investing in activities leading to the development of new antimicrobial agents. Financial support can be provided by government organizations, academic spin-outs, and foundations. Initiatives such as Combating Antibiotic Resistance Bacteria Biopharmaceutical Accelerator (CARB-X), *Global Antibiotic Research and Development Partnership* (GARDP), and Innovative Medicines Initiative (IMI) aim to fund the optimization of research through to clinical application. The goal of the next USD 1 billion Antimicrobial Resistance Action Fund, set up by the *International Federation of Pharmaceutical Manufacturers Associations (IFPMA)* and the WHO, is to bring two to four new antibiotics to market by 2030 [5].

The search for new phages with therapeutic potential against multidrug-resistant bacteria has become a promising alternative. Compared to the discovery, development, and production of antibiotics, therapeutic phages are easier to obtain and cheaper to produce [57]. Antibiotic resistance has contributed to the increasing use of combination or multidrug therapies involving the simultaneous use of at least two antimicrobial agents. Increased clinical efficacy may be provided by the simultaneous use of antibiotics and phages instead of sequential antibiotics. Combination therapy uses the beneficial effects of phages and antibiotics in the fight against bacterial infections. Phages have the ability to act precisely by infecting selected strains. They enable antibiotics to act by damaging bacterial cell walls, making bacteria more susceptible to phage attack. Phages remain insensitive to host specificity [58,59]. One potential limitation of phage therapy, especially monophage therapy, is the development of resistance. Therefore, phage cocktails are often proposed for phage therapy. Potential benefits of using phage cocktails include a broader spectrum of action and a reduced risk of bacterial resistance [1].

In 2005, a Phage Therapy Centre was set up in Wrocław to qualify patients for experimental phage therapy and to conduct it according to a protocol approved by the bioethics committee, as well as to monitor the patient’s condition after therapy. As no results from standard clinical trials of phage preparations are available to date, this form of therapy is currently only available as an experimental therapy.

## 3. *S. aureus* Phages

Phages are considered a promising alternative to rapidly emerging MDRB strains, such as methicillin-resistant *S. aureus* (MRSA), *S.haemolyticus* (MRSH), and *S.epidermidis* (MRSE) [60,61]. Staphylococcal phages have historically been classified into three families: Myoviridae, Siphoviridae, and Podoviridae [62]. Approximately 96% of the thousands of phages that have been visualized to date fall into these three categories [63]. The database contains 69 conserved genomes of virulent *S. aureus* phages, including 26 Podoviridae and 43 Myoviridae phages [2].

The diversity of the species *S. aureus* is mainly determined by mobile genetic elements, many of which are prophages or phage-associated genomic islands. In the vast majority of *S. aureus* strains, the presence of multiple bacteriophages in the genome has been confirmed, mediating niche adaptation by pathogens. Cell wall teichoic acid (WTA) acts as a common phage receptor for staphylococcal phages, and structural changes in WTA regulate phage–host specificity, thus shaping gene transfer between clonal lineages and even species [64]. Pathogenic staphylococci acquire resistance to many antibiotics mainly through generalized transduction [65]. Phage-mediated transduction enables the transfer of extrachromosomal mobile elements, such as plasmids, but also chromosomal markers. Transfer of chromosomal markers occurs with lower efficiency. Although some phages are able to transduce easily between staphylococcal species, horizontal gene transfer seems to be generally limited within a species by restriction-modification systems. The primary function of these systems is cell protection by degradation of foreign DNA [64]. In addition to the spread of antibiotic resistance genes via plasmid transfer, generalized transduction has recently been linked to the transmission of SCCmec cassettes associated with staphylococcal resistance to methicillin and other β-lactam antibiotics between strains of the same or different species [66,67].

## 4. Phage Therapies—Research

Examples of the successful use of phages against infections caused by *S. aureus* are well documented, particularly in cases of chronic infections of knee prostheses [68], infective endocarditis, heart valves [69], wound infections [70], and osteomyelitis [71]. Liu S. et al. presented the characterization of four lytic antitumor phages. These phages had a host range of over 98% of the MSSA and MRSA strains tested and strong antibiofilm properties. Good stability and infectivity profile were additional advantages of the phages tested. Given the complementary susceptibility profile of the phages and the reduced frequency of emergence of antibiotic resistance in the presence of the cocktail, the combination of the four phages in the APTC-C-SA01 cocktail suggests that the cocktail could be used for most *S. aureus* infections, including MRSA [1].

There is a lack of information in the literature on the penetration of staphylococcal phages in humans. Weber et al. presented the results of oral administration of staphylococcal phages [72]. Patients with purulent infections of Staphylococcus spp. etiology were treated with phages. After 10 days of treatment, positive results in 84% of serum samples and 35% of urine samples indicate high bioavailability of phages [73].

Literature data have demonstrated the efficacy of phage–antibiotic combination therapy against *S. aureus* in vitro infections. Kebriaei et al. [58] combined Sb-1 phage with daptomycin, resulting in a reduction of MRSA colonies in the biofilm. Simon et al. [74] showed in their studies a synergistic effect between phage Sb-1 and oxacillin. The use of phage Sb-1 alone contributed to a 35% reduction in the number of MRSA bacteria. The use of the phage in combination with oxacillin resulted in a 90% reduction in the number of MRSA bacteria. Joo et al. [75] also confirmed that the simultaneous use of vancomycin and bacteriophage K, compared with vancomycin or bacteriophage K used separately, reduced the number of MRSA bacteria in the biofilm. This observation indicates a significant synergistic effect of the antibiotic and bacteriophage. The results of laboratory studies, as well as clinical case reports, indicate a beneficial effect of phage therapy in patients with infections caused by drug-resistant *S. aureus*. Ferry et al. [68] presented the cases of three patients with recurrent prosthetic knee joint infections caused by the clinical isolate *S. aureus* (MSSA), who had significantly improved their clinical condition with the use of a phage cocktail in combination with antibiotics. In the first stage, patients were treated with antibiotics. Antibiotic therapy proved ineffective, and infections of the patient’s knee replacements were observed. In the next stage, three preparations of phage cocktails were prepared and injected directly into the patient’s joints, and combined antibiotic therapy lasting 6 to 12 weeks was used along with suppressive antibiotic therapy. Follow-up examinations were performed after 7, 11, and 30 months. In two patients, the results confirmed the presence of only mild synovial discharge. Synovitis was limited, and there were no persistent or recurrent bacterial infections. The patients showed a resolution of clinical symptoms and a decrease in the level of C-reactive protein. Another clinical case involves recurrent infection of a knee joint prosthesis with *S. aureus.* Ramirez Sanchez et al. [76] documented the successful intra-articular use of a phage cocktail and a single lytic phage in a 61-year-old female patient. Initially, intravenous cefazolin therapy combined with oral rifampicin and amoxicillin did not improve the clinical condition of the patient with knee joint infection. Clinicians used phage therapy based on intravenous infusions of the phage cocktail AB-SA01 (once every 12 h for 2 weeks) and cefazolin (once every 8 h for 6 weeks) [2].

Liu S. et al. [1] studied the isolated four *S. aureus* phages and tested them on 51 clinical MSSA and MRSA strains and reference strains. Phages were characterized by a wide host range, 82–94% individually and >98% in total, respectively. Such values significantly reduced the viability of *S. aureus* biofilms. Genome sequencing confirmed the lytic nature of all four phages. No resistance or virulence genes were identified. All phages showed high complementarity, with 49/51 strains (96%) susceptible to at least 2/4 of the tested phages. The combination of phages in the APTC-C-SA01 phage cocktail reduced the number of bacterial mutants produced that were insensitive to bacteriophages compared to therapy with phages used separately. The studies conducted confirmed that the APTC-CSA01 phage cocktail, comprising four lytic *S. aureus* phages, shows effective activity against MSSA and MRSA infections, which is the basis for further studies leading to its classification as a drug [1].

Phase I trials and clinical experience in the treatment of chronic rhinosinusitis (CRS) have been published in Eastern Europe, and their safety and efficacy have been reported [77]. CRS is a common problem with a prevalence of 1–15% worldwide [78]. It is defined as inflammation of the sinuses and nasal passages lasting more than 12 weeks. Anatomical, genetic, environmental, inflammatory, immunological, and microbiological factors are responsible for the occurrence of CRS [79]. Strains of bacteria that are difficult to treat are more frequently isolated from patients with CRS compared to patients with acute sinusitis. The presence of strains of *Staphylococcus* spp., *P. aeruginosa*, *Enterobacteriaceae* spp., anaerobes, and fungi has been confirmed. Treatment of patients with CRS includes a combination of antibiotics, local anti-inflammatories, antihistamines, immunological therapy, and functional endoscopic sinus surgery (FESS). Strains of *S. aureus* are commonly isolated from patients with CRS and cause difficulties in treatment. Antibiotic therapy and the FESS procedure do not result in recovery in many patients [80]. The role of biofilms in the pathogenesis of CRS needs further clarification [78]. Fong et al. [81] report that bacterial biofilms contribute to treatment failure in patients with CRS. Others, however, emphasize the importance of the beneficial role of certain biofilms in sinus physiology, particularly the role of commensal bacteria [78]. Treatment-resistant CRS is most often associated with the presence of biofilms formed simultaneously by multiple bacterial species rather than a single species. The presence of *S. aureus* and *P. aeruginosa* was detected in severe stages of sinus infection, while biofilms of *H. influenzae* were associated with mild symptoms or favorable outcomes after surgery [78]. Phages (mainly phage cocktails) or lysins can be considered as alternatives to antibiotics in the treatment of CRS infections accompanied by biofilm formation. Rodriguez M. et al. [77] presented the case of a patient who showed signs of disease and positive microbiological tests for many years despite multiple operations and antibiotics. The patient was successfully treated. The systemic application of phage therapy helped to a limited extent, and significant improvement was observed only after frequent topical administration, probably due to the achievement of higher local phage concentrations. It is also possible that the systemic immune response is reduced after intravenous administration.

Szaleniec et al. [82] investigated the activity of phage cocktails of *S. aureus* and *P. aeruginosa* against bacterial isolates from patients with acute exacerbation of CRS. The researchers showed that 59% of pathogens, including 81% of antibiotic-resistant pathogens, were sensitive to phages from the Biophage Pharma collection; 63% of *S. aureus* isolates and 40% of *P. aeruginosa* isolates were characterized by sensitivity to phage cocktails. Zhang et al. [83] tested the ability of *S. aureus* phage to effectively kill antibiotic-resistant clinical strains isolated from CRS patients. The study used two virulent *S. aureus* phages Sa83 and Sa87 from the family Myoviridae (AmpliPhi, Australia) and 65 MDR antibiotic-resistant clinical strains (Cls) (resistant to three or more classes of antibiotics) isolated from CRS patients; 71.1 percent (42 of 59) and 69.4 percent (41 of 59) of *S. aureus* strains resistant to antibiotics were sensitive to Sa87 and Sa83 phages, respectively. A summary of phage studies is presented in Table 1.

## 5. Benefits of Phage Therapy

In the course of evolution, bacteriophages have adapted to penetrate biofilms and infect bacterial cells hidden inside complex bacteriophage structures [11]. Bacteriophages have the property of self-replication, which favors an increase in the concentration of phage particles at the site of infection, in contrast to antibiotics, which show a gradual decrease in concentration after application [8]. Phages are characterized by a high specificity towards a single bacterial species, which minimizes disruption of the coexistence of commensal bacteria and the normal microflora, as well as phage self-replication at the site of infection. Bacteria, as with antibiotics, can gain resistance to phages in the course of evolution, which results in the ineffectiveness of phage therapy. The use of phage cocktails reduces the tendency of bacteria to develop resistance to phages present in the cocktail. By using several phages targeting the same bacterial species, it is possible to expand the range of hosts, which significantly increases the possibilities of using phages in a clinical context [1].

The wide distribution of bacteriophages in the natural environment allows for a quick and inexpensive search for antimicrobial agents that are an alternative to expensive research on new antibiotics. Phages are characterized by a high level of specificity towards bacteria, destroying only pathogenic bacteria and leaving the physiological flora intact. Broad-spectrum antibiotics can disturb the balance of bacterial flora. The number of phages naturally decreases after the removal of host bacteria after eliminating the target bacteria in vivo, preventing the accumulation of bacteria and the associated toxic effects and side effects. Phages selectively infect bacteria without entering human cells and without disturbing their normal metabolism. No cases of mutagenic or genotoxic effects of phages on the human body have been observed [2].

The generation of resistance to phages is 10 times slower than the generation of resistance to antibiotics [84]. The clinical importance of bacteriophages results from the possibility of combining them with antibiotics without causing bacterial resistance to antibiotics. Phages can also disrupt the structure of bacterial biofilm by secreting cell wall-degrading hydrolases and extracellular polysaccharide depolymerase [2]. The safety of phage therapy requires confirmation by a larger number of studies.

## 6. Limits of Therapy

Phage therapy, despite the growing number of experiments confirming the possibility of using virulent phages as therapeutic agents, is still in the early stages of development. Unlike antibiotic therapy, new factors must be taken into account when developing phage therapy, including the multiplicity of infections, the size and availability of the target bacterial population, and the rate of acquiring bacterial resistance in response to phage pressure. In addition, the limited understanding of the molecular mechanisms underlying the physiology of phages during interactions with bacterial cells requires attention and research. The use of transcriptomic analysis may be beneficial in addressing this issue by studying gene expression in the phage–bacteria system, which is key to understanding gene function during specific stages of infection [85].

Pathogenic bacterial strains are used during the amplification of therapeutic phages; therefore, the production of virulence factors by bacteria and infection of the phage may contribute to the occurrence of undesirable reactions during the administration of phages to patients. Prophages, which are genomes of lysogenic phages integrated into the bacterial chromosome of the host, are often an element of bacterial genomes. Prophages can be excised and then enter the lytic cycle and produce new virions as a consequence of stress caused by infection with lytic phage. There is a risk of integration of prophages into the genome of the patient’s bacterial strain (lysogenization) and possible transfer of virulence and resistance genes or modulation of virulence gene expression and/or interaction with the patient’s organism. Research related to the development of therapeutic phage purification and quality control processes is needed [40]. Only lytic phages can be used safely in the clinical setting. Furthermore, it must be shown that phages that are candidates for use are lytic and lack lysogenic genes that would enable them to integrate their genetic material into the bacterial host [86,87].

The predictability and reliability of treatment results depend to a large extent on complex and not fully understood interactions between phages, antibiotics, and bacteria. In addition, the increasing resistance of bacteria to phages has a negative impact on the effectiveness of phage therapy. Combining antibiotics and phages and using their synergistic properties aims to prevent the emergence of bacterial resistance to both species. In the initial stage of infection, phages are adsorbed to cellular receptors. Some bacteria have developed mechanisms that prevent this key process [88]. The use of phage cocktails and the administration of an increased number of phages in the initial stage of treatment have a decisive impact on reducing the risk of bacterial resistance to phages [89]. Phages can have difficulty penetrating and distributing evenly in biofilms. This process reduces phage proliferation, resulting in reduced treatment efficacy [90]. The difficulty in using phage therapy is the lack of defined treatment regimens, including the lack of standardization regarding dosage, route of administration, and the number of phages used [78]. More clinical trials are needed to facilitate the development of phages in clinical settings. Available treatments are based on case reports or small series of clinical trials. Knowledge about the in vivo activity of phages is limited despite the recognition of their antimicrobial properties in vitro. More data are also needed for the use of phages in patients in healthcare facilities [7].

Phages are characterized by a narrow host range. Bacteria rapidly develop resistance to phages. So, it is necessary to use complex cocktails containing phages with different mechanisms of operation. This practice poses a risk of lack of safety due to the large number of unidentified genes in phage genomes that have the ability to promote bacterial virulence or cause undesirable side effects. High doses of phages may be responsible for the occurrence of an undesirable immune response. What is more, the pharmacodynamics and pharmacokinetics of phages in the human body require further investigation [91]. In addition to medical implications, practical considerations are also important. For example, the need to purify therapeutic phages in accordance with current good manufacturing practices is crucial. Such efforts impact the economic feasibility of large-scale phage production [92]. There are numerous regulatory hurdles that need to be addressed, resulting from differences in the approval of conventional antimicrobials and phage therapies. Routine implementation of phage therapy requires the involvement of multiple sectors in interdisciplinary collaboration, including academia, industry, and governance entities [65].

The ability of phages to acquire, transfer, and transmit bacterial gene fragments poses a risk of spreading drug-resistant genes [93]. The limitation of the wide use of phage preparations and the narrowing of their range of action is largely due to the high degree of specificity of phages in recognizing and infecting host bacteria. As in the case of antibiotics, repeated use of phage preparations can lead to bacterial tolerance to phage infection. The use of phages is not associated with the occurrence of toxicity or side effects; however, during the lysis of Gram-negative bacteria, phages can release endotoxins, reducing the safety of therapy [94].

Clinical evidence supporting the efficacy of phage therapy is insufficient. The independent effect of phage therapy is often difficult to assess due to concomitant combination therapy. Insufficient knowledge of phages and phage therapy among clinicians, patients, and their families can often lead to inappropriate implementation of therapy. The effectiveness of phages in combating systemic infections can be increased by reducing phage immunogenicity through masking or modifying antigenic epitopes. For this purpose, phage encapsulation with liposomes, nanomaterials, or hyaluronic acid is used [95].

## 7. Bacterial Lysins (Endolysins)

Bacteriophages as a whole, as well as factors isolated from them, such as specific enzymes (endolysins) involved in the rapid degradation of the cell wall, may be responsible for the destruction of bacterial cells. Endolysins are enzymes synthesized at the last stage of the phage replication cycle. They are characterized by high specificity in the degradation of the peptidoglycan layer of bacterial cell walls. Due to this property, bacteriophages can lyse and degrade bacterial cells with high efficiency without affecting human cells. Literature data have confirmed the specificity of phage endolysins against *Gardnerella* strains [96], *E. faecalis* [97], *A. baumannii*, *P. aeruginosa* [98], and *S. aureus* [99]. Endolysins also showed the ability to remove biofilm. For example, the endolysin LysP108 revealed a 90% antibacterial potential against MRSA. Endolysins are extremely promising as a new antimicrobial class. The interest in bacteriophages and their use in clinical practice are evidence of innovative solutions used in modern science aimed at increasing the effectiveness of the treatment of MRSA infections [7].

The widespread use of phage therapy using complete phages is an alternative to antibiotic therapies, but it is characterized by a limited antibacterial spectrum, complex preclinical and clinical evaluations, and a lack of a sufficient regulatory framework. Lysines have proven to be an alternative to antibiotics. The main advantages of using lysines include: no proliferation, high bactericidal activity, broad host spectrum, well-defined pharmacokinetics, reduced likelihood of resistance, and antibody development. The effectiveness of lysines is largely influenced by the presence of the outer membrane in Gram-negative bacteria, which is a physical barrier to lysines. Lysines can, however, directly degrade peptidoglycan bonds, leading to lysis of the cell wall of Gram-positive bacteria. This property of lysines confirms their clinical efficacy in infections caused by Gram-positive bacteria. In 1959, a lysine with bactericidal properties was successfully purified. In 2001, phage lysins were first used as topical antimicrobial agents [2].

Lysines have many advantages over classical antibiotics: minimal off-site invasiveness, preventing significant disruption of the microbiome, rapid bactericidal activity, and efficacy against multidrug-resistant bacteria and biofilms. Due to the synergy of some lysines with clinically relevant antibiotics, their potential for therapeutic use is much greater. The main limitations of the use of lysines include their short half-life in plasma, potential immunogenicity, and lack of efficacy in combating intracellular bacteria [100]. SAL200 (Tonabacase) is the first lysine tested in humans in a phase I clinical trial. SAL200 is a recombinant derivative of SAL-1 lysine produced by the staphylococcal bacteriophage SAP-1. Data from the literature confirm that SAL200 demonstrates bactericidal efficacy against capsulated strains of MRSA and MSSA with the ability to form biofilm. SAL200 lysine retains its activity in vivo during the treatment of patients with systemic *S. aureus* infection, as well as in the case of pneumonia in a mouse model of infection. A significant prolongation of patient survival has been documented [101,102]. The safety and tolerability of lysine in the Phase I study provided the basis for conducting a Phase II study in patients with chronic *S. aureus* bacteremia (NCT03089697). The FDA also approved a US-led Phase II study for the treatment of complicated *S. aureus* bacteremia and infective endocarditis (Tonabacase—iNtRon Biotechnology, Basilea Pharmaceutica Ltd, Allschwil, Switzerland) [5].

## 8. Antimicrobial Peptides

Antimicrobial peptides are referred to as new antibiotics. They have antibacterial, immunoregulatory, and antimicrobial effects. They are found in almost all life forms. In vertebrate organisms, they can be found in white blood cells and epithelial surface cells, including in the oropharyngeal region, lungs, and nasal cavity [103].

Antimicrobial peptides (AMPs) have a high potential to combat antibiotic-resistant bacteria, which is why they are a promising alternative to conventional antibiotics. The increased bactericidal efficacy of AMPs compared to antibiotics is supported by their pharmacodynamics [104]. AMPs are naturally produced small molecules that are part of the innate immune system of various organisms [105]. Despite the great diversity of AMPs, these molecules are also characterized by common features, such as 3D structures [106]. The diversity of AMPs is mainly due to different amino acid content, activity, mechanisms of action, origin, and physicochemical properties [107]. The criterion for dividing AMPs into antibacterial, antifungal, antiviral, and antiparasitic peptides is their activity [108]. Antibacterial AMPs are characterized by a diverse range of mechanisms and targets of action. AMPs are responsible for inducing damage to the bacterial membrane or inhibiting the synthesis of proteins, enzymes, and nucleic acids at the cytoplasmic level [109]. The above-mentioned features of AMPs constitute the basis for classifying AMPs as agents with a high potential for combating bacteria sensitive and resistant to conventional antibiotics. Literature data confirm the efficacy of individual AMPs in combating bacterial infections, including VRSA strains and *S. aureus* strains with reduced susceptibility to vancomycin. There is ongoing research into the activity of peptides, their possible molecular targets, and mechanisms of action [110].

Some antimicrobial peptides may exhibit hemolytic or cytotoxic effects. Several methods of designing artificial AMPs have been implemented to reduce side effects and increase antimicrobial activity. Epidemiological surveillance of resistant strains has been greatly facilitated by advances in the discovery, design, optimization, synthesis, and evaluation of new antimicrobial peptides, both natural and artificial. In the clinical practice of treating infectious diseases, great potential is attributed to the use of AMPs with potent and rapid antimicrobial activity and non-toxicity to human cells [110].

Defensins and cathelicidins are common groups of antimicrobial peptides [111]. In humans, beta-defensins 1–4 (HBD 1–4) have been confirmed. HBD-1 is continuously expressed. Proinflammatory stimuli induce the expression of HBD 2-4 [112]. Human beta-defensin 2 (HBD-2), secreted by epithelial cells, has antimicrobial activity and the ability to stimulate the expression of Toll-like receptors on the surface of immune cells and their chemotaxis. HBD-2 plays an important defensive role in the oral cavity. It has been shown that in the absence of inflammation, gingival tissue secretes HBD-2 continuously. Deficiency of HBD-1 and HBD-2 in the nasal epithelium may contribute to chronic rhinosinusitis [103].

Human beta-defensin 3 (HBD-3) also exhibits antimicrobial activity and is associated with chemotaxis. HBD-3 has demonstrated broad-spectrum activity against several drug-resistant bacteria [10]. This peptide is found in epithelial cells of the respiratory tract, genitourinary tract, stomach, and intestine, as well as in non-epithelial tissues. HBD-3 has demonstrated efficacy in eliminating biofilms of *Staphylococcus* antibiotic-resistant strains [113]. HBD-3 also showed strong activity against Gram-negative bacteria such as *P. aeruginosa*, *A. baumannii*, and *Stenotrophomonas maltophilia*, among others, with bactericidal activity within 1–5 min [103].

Catelicidin LL 37 (LL 37) is a basic defense mechanism against bacteria and other pathogens in the inflammatory state. Cathelicidins have the ability to destroy biofilms, viruses, parasites, and fungi. They are also characterized by the ability to modulate and stimulate cells of the innate and specific immune system. The use of cathelicidin LL 37 has been presented in detail in patients with chronic rhinosinusitis. Its expression is observed in the nasal mucosa in a healthy state, with a significant increase during inflammation [114]. LL 37 levels also appear to be increased in CRS patients in response to fungal pathogens [115]. Reduced epithelial levels of HBD-2, HBD-3, and LL 37 suggest epithelial barrier dysfunction in patients with nasal polyps [103].

Bacterial AMPs are also known as bacteriocins. Nisin A is a bacteriocin produced by *Lacobacillus lactis* [116]. Due to its broad spectrum of antimicrobial activity, mainly against Gram-positive bacteria [117,118], nisin A is widely used worldwide as a food additive to prevent food poisoning [119]. In addition, Alves et al. described the potential use of nisin in combination with oxacillin for the treatment of methicillin-resistant *S. aureus* [120]. The characteristics of the most commonly used antimicrobial peptides are presented in Table 2.

## 9. Vaccines

A vaccine against *S. aureus* could help prevent colonization, reduce the incidence of infections caused by this microorganism, reduce mortality associated with severe infections, especially in children and the elderly, and reduce hospitalization expenses for patients [10].

To date, no vaccine has been approved against *S. aureus* strains [121,122]. Different structures of *S. aureus* were the target of the study, such as surface polysaccharide (poly-N-acetylglucosamine) [123], surface proteins such as the iron-regulated surface proteins IsdA or IsdB [124], Clumping Factor A *(Clf) A or ClfB* [125] *and fibronectin-binding protein (FnBP)* [126]. The vaccines tested showed only partial efficacy [127]. These factors are not essential for bacterial life, which may explain the ineffectiveness of vaccines that take these factors into account [128]. Studies combining multiple antigens in a single vaccine have also been conducted. Recent studies target *S. aureus* molecules that stimulate an immune response similar to that observed during natural *S. aureus* infection. Some studies suggest a lack of proper action in humans despite promising results in animal models [127].

The difficulty in selecting universal strains for use in vaccines is the high diversity of *S. aureus* strains, their high virulence, their rapid adaptation to changing environmental conditions, and the very high diversity of target populations and the range of diseases with *S. aureus* etiology. Producing a universal vaccine appears to be a daunting task, so researchers’ efforts are currently focused on developing vaccines against a specific *S. aureus* infection or group of them. Vaccine research has also brought other benefits, such as a large number of vaccine platforms, including a variety of antigens, new adjuvants, and delivery systems. However, further research is needed.

## 10. Natural Methods

There is an urgent need to discover new alternatives to antibiotics in the fight against infections. In addition to research involving antimicrobial peptides, vaccines, and receptors that recognize molecular patterns, researchers’ efforts are also focusing on the potential use of plant extracts, honey, and propolis [2].

Nature is an invaluable source of bioactive molecules characterized by great chemical diversity. They may provide a unique platform for providing new scaffolds for further chemical modification to obtain compounds with optimized biological activity. Alkaloids, which are important secondary metabolites produced by a large number of organisms, including bacteria, fungi, plants, and animals, are natural compounds with diverse biological activities. Studies on the evaluation of the antimicrobial activity of naturally occurring alkaloids began as early as 1940. They led to the identification of several potent monomeric and dimeric alkaloids. Synthetic modifications have been successfully introduced to improve the biological activity of alkaloids [129,130]. However, there is an unusually large discrepancy between their historical importance and their role in the development of modern drugs. No alkaloids that are antimicrobial drugs are available on the market [131]. Casciaro B. et al. [132] used an internal library of approximately 1000 natural products to identify new potential antimicrobial alkaloids. Based on the screening of all alkaloids present in this library, a rare β-carboline heterodimer, nigritanin, was identified, which showed strong anti-staphylococcal activity. Nigritanin isolated from Strychnos nigritana was characterized by its antimicrobial activity against clinical isolates of *S. aureus*. Its potential cytotoxicity in the short and long term was also assessed. Nigritanin showed remarkable antimicrobial activity without being toxic in vitro to cells. Given these interesting results, nigritanin can be considered a promising candidate for the development of new antimicrobial molecules for the treatment of infections caused by *S. aureus*.

Peptoids show potential as novel therapeutic agents. Low concentrations of peptoids are sufficient to kill both MSSA and MRSA strains. Peptoid 1, in particular, is effective, but so is Peptoid 1-C134mer. Benjamin et al. conducted experiments to determine the ability of peptoids to prevent biofilm formation and the ability to remove biofilms. Peptoid 1 has shown effective activity, probably related to its ability to kill *S. aureus* bacteria in dispersed form and to its interaction with biofilm matrix elements [133].

Manuka honey has significant antibiofilm activity in vitro and in vivo against *S. aureus*, methicillin-resistant *S. aureus*, and *P. aeruginosa.* Ooi et al. [134] conducted the first randomized phase I clinical trial evaluating the safety and efficacy of Manuka honey in combination with methylglyoxal irrigations in patients with antibiotic-resistant chronic rhinosinusitis. The authors included patients who had previously undergone endoscopic sinus surgery and presented with physical and subjective symptoms of sinus infection. Bacterial cultures were cultured from swabs taken from the sinuses. Patients were randomly allocated to a group using Manuka honey with methylglyoxal or to a group using saline solution and antibiotic therapy. The results indicate that the use of Manuka honey in combination with methylglyoxal is safe, and the efficacy is comparable to but not better than the use of saline irrigation in combination with antibiotic therapy.

Research using essential oils to treat patients with bacterial infections is ongoing. Essential oils have a broad spectrum of antimicrobial activity against many foodborne pathogens, including Gram-positive and Gram-negative micro-organisms. Gram-positive bacteria are more sensitive to oils than Gram-negative bacteria [135]. Plant essential oils such as clove, cinnamon, oregano, and thyme oils are among the most promising antibacterial alternatives, showing potential for treating *S. aureus* infections and minimal chance of developing resistance. However, their direct use or incorporation into disinfectants or medicinal preparations is ruled out due to their low solubility, chemical instability, high reactivity, and potential for toxicity/immunogenicity to human cells. Ivanova et al. used nanocapsules enriched with antibacterial oregano essential oil. The preparation inhibited the growth of *S. aureus*. It may, therefore, represent an effective therapeutic method in *S.aureus* infections to avoid the use of antibiotics and prevent the development of antimicrobial resistance [136]. Literature data indicate that the oil obtained from Schinus areira L. has shown antimicrobial activity against various bacteria, including *S. aureus*. Benbelaïd et al. reported that an essential oil extracted from the leaves and fruits of *S. areira*, obtained from samples collected in Jujuy (Argentina), showed antimicrobial activity against a methicillin-resistant strain of *S. aureus* [137]. Although, to our knowledge, there is extensive documentation on the antimicrobial properties of *S. areira* essential oil against *S. aureus* and other bacteria, studies focusing on the possible mechanism of action as well as bacterial targets are still scarce. Further, more extensive research is needed.

## 11. Further Developments

The main obstacle to the modernization of phage therapy is the high level of specificity of phages for their bacterial hosts. In order to cover a wide spectrum of pathogenic bacteria, it is necessary to use a large number of phages. Personalized phage therapy provides individual preparation of a therapeutic phage cocktail for individual patients based on the sensitivity of bacteria to phages. Selection of appropriate phages for therapeutic use requires screening tests of bacterial susceptibility to phages in order to select the fastest possible identification of infectious phages from a large collection of phages [138]. Rapid diagnostic tests assessing phage susceptibility are, therefore, an emerging branch of interest for researchers and clinicians, with the main challenge being the establishment of standardized methods to identify suitable phages for personalized phage therapy [139].

Phage engineering technology enables the design of phage receptors in highly conserved regions of pathogenic bacteria. Another application of genetic engineering involves the modification of phage receptor binding proteins, which results in the expansion of their host range [139]. Updating regulatory strategies to take into account the complexity of phage products is essential for the standardization of phage preparations for clinical treatment [3]. Difficulties in selecting optimal conditions for phage therapy (time, dosage, routes of administration) result from the unique features of phage self-replication and growth in clinical conditions. Extensive studies are needed to clarify the pharmacokinetic principles of phage self-replication and growth [140]. This area is a major topic for further phage research.

Phage therapy in the treatment of patients with MRSA infections is a promising alternative to antibiotics in times of increasing antibiotic resistance. Using the specificity of bacteriophages towards their bacterial hosts, an innovative approach to targeted therapy is possible, minimizing the impact of phages on the human microbiome and the occurrence of resistance to broad-spectrum antibiotics [141]. Incorporating advanced phage engineering techniques, such as CRISPR-Cas systems, into the therapeutic process can significantly increase the effectiveness and specificity of phage therapies. A key topic of future research is the development of rapid and precise diagnostic tools for the identification of bacterial strains, which is the basis for implementation in the next stage of phage therapy [142]. Currently, research should focus on bridging the gap between in vitro efficacy and clinical application, which will translate into laboratory success into measurable clinical results. A multidisciplinary approach integrating molecular biology and medicine is necessary to fully utilize the therapeutic potential of bacteriophages [7]. There is an urgent need to investigate the diversity and impact of lytic staphylococcal phages and all varieties infecting species other than *S. aureus*. Furthermore, as a complement to the focus on phages themselves, immune systems directed against them should be considered, as the latter are powerful forces driving phage evolution and also influence the possible reduction in efficacy of phage-based antimicrobials [65]. Antimicrobial peptides, as well as antimicrobial agents of natural origin, including essential oils and peptoids, used as antibiotic alternatives, also require research that considers a wider range of patients.

## Figures and Tables

**Table 1 pharmaceuticals-18-00081-t001:** Phage research summary table.

Authors	Phage Type	Target Strains	Additional Information
Liu et al. [1]	APTC-C-SA01	*Staphylococci*, MRSA, MSSA	Lytic phages
Weber et al. [72]	Phage cocktail	Staphylococci	Orally administered
Kebriaei et al. [58]	Sb-1 phage with daptomycin	MRSA, biofilm	
Simon et al. [74]	Phage Sb-1 and oxacillin	MRSA	
Joo et al. [75]	Combined use of vancomycin and bacteriophage K	MRSA, biofilm	
Ramirez-Sanchez et al. [75]	AB-SA01, phage cocktail and cefazolin	MSSA	Patient’s knee joint infection
Szaleniec et al. [82]	Bacteriophages from the Biophage Pharma collection	MRSA, MSSA, *coagulase-negative staphylococci*, *P. aeruginosa*	
Zhang et al. [83]	Sa83 and Sa87 from the family Myoviridae (AmpliPhi, Australia)	*S. aureus*	CRS patients

**Table 2 pharmaceuticals-18-00081-t002:** Antimicrobial peptides summary table.

Name	Characteristics	Additional Information
Human beta-defensins 1–4 (HBD 1–4)
HBD-1	continuously expressed	Deficiency of HBD-1 and HBD-2 in the nasal epithelium may contribute to chronic rhinosinusitis
HBD-2	-secreted by epithelial cells,-antimicrobial activity,-ability to stimulate the expression of Toll-like receptors on the surface of immune cells and their chemotaxis,-an important defensive role in the oral cavity
HBD-3	-antimicrobial activity,-associated with chemotaxis,-broad-spectrum activity against several drug-resistant bacteria,-efficacy in eliminating biofilms of *Staphylococcus* antibiotic-resistant strains,-strong activity against Gram-negative bacteria such as *P. aeruginosa*, *A. baumannii, Stenotrophomonas maltophilia*,-found in epithelial cells of the respiratory tract, genitourinary tract, stomach, and intestine, as well as in non-epithelial tissues.	Bactericidal activity within 1–5 min
HBD 2–4	-proinflammatory stimuli induce the expression of HBD 2–4.	
Cathelicidin LL 37	-defense mechanism against bacteria and other pathogens in the inflammatory state,-ability to destroy biofilms, viruses, parasites and fungi,-ability to modulate and stimulate cells of the innate and specific immune system,-cathelicidin LL 37 expression is observed in the nasal mucosa in a healthy state, with a significant increase during inflammation,-LL 37 levels also appear to be increased in chronic sinusitis patients in response to fungal pathogens	
Nisin A	-a bacteriocin produced by *Lacobacillus lactis,*-broad spectrum of antimicrobial activity, mainly against Gram-positive bacteria,-nisin A is widely used worldwide as a food additive to prevent food poisoning,-potential use of nisin in combination with oxacillin for the treatment of methicillin-resistant *S. aureus.*

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
