# Peer review of "Antibiotic Resistance of Staphylococcus aureus Strains—Searching for New Antimicrobial Agents—Review"

_pharmaceuticals, 2025, doi:10.3390/ph18010081_

Round 1

Reviewer 1 Report

Comments and Suggestions for Authors

1.       Authors should revise the introduction with up-to-date and relevant statistics on AMR, which would help emphasize the growing global threat of AMR and make the introduction more compelling and underscore the urgency of the research.

2.       The authors provided a comprehensive discussion on phage therapy, but the information on other strategies to combat S. aureus inhibition, such as endolysins, AMPs, natural sources, and vaccines, is relatively brief. It is recommended that the authors expand this section to provide a more detailed review of these alternatives, offering a balanced overview of the various therapeutic strategies available.

3.       According to scientific guidelines, organism/bacteria names should be italicized throughout the manuscript. Authors should follow this formatting for consistency and adherence to scientific standards?

4.       The manuscript exhibits inconsistencies in font size and style (359, 588, 618, and 635). The authors should ensure uniformity in the font size and style throughout the manuscript, adhering to the journal's formatting guidelines.

5.       Certain sections of the manuscript contain grammatical, typographical errors (Eg: line 32, 120 121, 322) and exhibit poor sentence formation. It is recommended to carefully revise the entire manuscript for smoother readability?

Comments on the Quality of English Language

The manuscript should undergo a thorough review to correct any grammatical and typographical errors to enhance the readability and professionalism of the manuscript

Author Response

Comments 1: Authors should revise the introduction with up-to-date and relevant statistics on AMR, which would help emphasize the growing global threat of AMR and make the introduction more compelling and underscore the urgency of the research.

Thank you very much for your attention. Information regarding AMR is included in the manuscript text: It is predicted that by 2050, infections associated with antimicrobial resistant bacteria (multi drug resisistant bacteria, MDRB) will be associated with approximately 10 million deaths per year, surpassing the mortality attributed to cancer (page number 2)

Comment 2: The authors provided a comprehensive discussion on phage therapy, but the information on other strategies to combat S. aureus inhibition, such as endolysins, AMPs, natural sources, and vaccines, is relatively brief. It is recommended that the authors expand this section to provide a more detailed review of these alternatives, offering a balanced overview of the various therapeutic strategies available.

The topic was partially discussed in the manuscript. The author's goal was to focus on phage therapy, mentioning only other antimicrobial therapies.

Comment 3: According to scientific guidelines, organism/bacteria names should be italicized throughout the manuscript. Authors should follow this formatting for consistency and adherence to scientific standards?

Thank you for pointing this out. Changes have been made to the manuscript.

Comment 4: The manuscript exhibits inconsistencies in font size and style (359, 588, 618, and 635). The authors should ensure uniformity in the font size and style throughout the manuscript, adhering to the journal's formatting guidelines.

Thank you for pointing this out. Changes have been made to the manuscript.

Comment 5: Certain sections of the manuscript contain grammatical, typographical errors (Eg: line 32, 120 121, 322) and exhibit poor sentence formation. It is recommended to carefully revise the entire manuscript for smoother readability?

Thank you for pointing this out. Changes have been made to the manuscript.

Reviewer 2 Report

Comments and Suggestions for Authors

The review focuses on combating Staphylococcus aureus infections, especially methicillin-resistant strains (MRSA), amidst rising antimicrobial resistance. It highlights phage therapy as a promising alternative, alongside other options like endolysins, antimicrobial peptides, plant extracts, and essential oils. A multidisciplinary approach is emphasized to advance these therapies and address challenges such as safety standards and bacterial resistance mechanisms.

Comments:

1. Line 303-315: Could authors compare the therapies used combined antibiotics and combined phages? Did they share the similar advantages and limits?

2. Line 348-442: Could author make a table to summarize all these research (e.g. phage type, target strain, killing effect.)?

3. Line 587-648: AMP is a large family, could author make a table to summarize the significant research and commercial products?

4. Could author also discuss the future development for other therapies mentioned in manuscript (except phage)?

5. Line 76-81: Please provide references.

6. Line 98-100: Please provide references.

7. Line 113-122: Please provide more references. And why did author use bold & larger font for some paragraphs?

8. Line 472: This title is very confusing, how about using "limits of therapy"?

9. Line 328-330: please fix the format error.

Author Response

Comment 1. Line 303-315: Could authors compare the therapies used combined antibiotics and combined phages? Did they share the similar advantages and limits?

Comparison presented in the manuscript text

Comment 2: Line 348-442: Could author make a table to summarize all these research (e.g. phage type, target strain, killing effect.)?

Thank you for pointing this out. Changes made to the manuscript

Comment 3: Line 587-648: AMP is a large family, could author make a table to summarize the significant research and commercial products?

Thank you for pointing this out. Changes made to the manuscript

Comment 4: Could author also discuss the future development for other therapies mentioned in manuscript (except phage)?

The topic was partially discussed in the manuscript. The author's goal was to focus on phage therapy, mentioning only other antimicrobial therapies.

Comment 5:   Line 76-81: Please provide references.

Álvarez, A.; Fernández, L.; Gutiérrez, D. et al. Methicillin-Resistant Staphylococcus aureus in Hospitals: Latest Trends and Treatments Based on Bacteriophages. J Clin Microbiol 2019; 57(12):e01006-19 

Comment 6: Line 98-100: Please provide references.

Álvarez, A.; Fernández, L.; Gutiérrez, D. et al. Methicillin-Resistant Staphylococcus aureus in Hospitals: Latest Trends and Treatments Based on Bacteriophages. J Clin Microbiol 2019; 57(12):e01006-19 

Comment 7 Line 113-122: Please provide more references. And why did author use bold & larger font for some paragraphs?

Parts of the text were mistakenly underlined.

F. Maurice, I. Broutin, I. Podglajen, P. Benas, E. Collatz, F. Dardel; Enzyme structural plasticity and the emergence of broad-spectrum antibiotic resistance; EMBO Rep. (2008), 10.1038/embor.2008.9

Comment 8: Line 472: This title is very confusing, how about using "limits of therapy"?

I agree with the reviewer. The change has been made in the manuscript.

Comment 9: Line 328-330: please fix the format error.

Thank you for pointing this out. The change was made in the manuscript.